# Heterogeneous Subgraph Transformer for Fake News Detection

## ABSTRACT

Fake news is pervasive on social media, inflicting substantial harm on public discourse and societal well-being. We investigate the explicit structural information and textual features of news pieces by constructing a heterogeneous graph with regard to the relations among news topics, entities, and content. Through our study, we reveal that fake news can be effectively detected in terms of the atypical heterogeneous subgraphs centered on them. These subgraphs encapsulate the essential semantics of news articles as well as the intricate relations between different news articles, topics, and entities. However, suffering from the heterogeneity of topics, entities, and news content, exploring such heterogeneous subgraphs remains an open problem. To bridge the gap, this work proposes a hierarchical framework - heterogeneous subgraph transformer (HETEROSGT) - to exploit subgraphs in our constructed heterogeneous graph. In HETEROSGT, we first apply a pre-trained dual-attention language model to derive textual features in accordance with word-level and sentence-level semantics. Then, we employ random walk with restart (RWR) to extract subgraphs centered on each news. The extracted subgraphs are further fed to our proposed subgraph Transformer to encode the subgraph surrounding each news piece for quantifying its authenticity. Extensive experiments on five real-world datasets demonstrate the superior performance of HETEROSGT over five baselines. Further case and ablation studies validate our motivation in investigating the subgraphs centered on news and demonstrate that performance improvement stems from our specially designed components. The source code of HETEROSGT is available at https://github.com/HeteroSGT/HeteroSGT.

## CCS CONCEPTS

• **Computing methodologies** → **Natural language processing**;
• **Information systems** → **Web searching and information discovery**; **Data mining**.

## KEYWORDS

Fake news detection, misinformation and disinformation, Transformer, heterogeneous graph

**ACM Reference Format:**
Anonymous Authors. 2018. Heterogeneous Subgraph Transformer for Fake News Detection. In *Proceedings of Make sure to enter the correct conference title from your rights confirmation emai (Conference acronym 'XX)*. ACM, New York, NY, USA, 9 pages. https://doi.org/XXXXXXX.XXXXXXX

## 1 INTRODUCTION

From a recent survey by Reuter, merely 42% of users, on average, place trust in the news they encounter online most of the time[1]. The limited trust can be attributed to the immense volume of news articles online accompanied by an escalating prevalence of fake news [5, 43, 44], which pervades multiple domains, spanning politics, economics, health, and beyond. Such deceptive content inflicts substantial and lasting harm to the public interest and social well-being. For instance, the fake news claiming that '5G technology can spread coronavirus' led to over 20 mobile phone masts in the UK being vandalized[2].

Regarding the deceitful content of fake news, extensive research efforts have been devoted to the exploration of text content in each news article to mitigate the detrimental consequences. These content-based language methods typically focus on the textual features associated with the news articles, including the linguistic features, syntactic features, writing styles, and emotional signals [10, 24]. But when dealing with well-crafted fake news that closely mimics real news in terms of textual features, these methods suffer from the low discriminativeness between the fake news and genuine information, leading to compromised performance.

Others mitigate the indistinguishable textural features of fake news by investigating additional structural information from the news parse trees [1, 4, 42], which delve into the relations among words, phrases, and sentences. Their improved performance can be primarily attributed to the exploration of the grammatical structure of news content in a hierarchical manner. However, the higher-level knowledge encapsulated in the relations among news articles[3], news entities, and topics still lacks sufficient exploration. More recent works even take account of the news dissemination in online social networks and detect fake news with regard to their propagation patterns [8, 21, 23, 30]. Due to their dependency on the additional social structure and monitoring the message dissemination process, these methods fall short of the real-time detection of fake news and are severely tackled by the large scale of social networks.

Instead, we consider exploiting the text content of news articles concerning both the rich grammatical patterns among words and sentences, as well as the complex relations among the news entities, topics, and news articles. All this information can be promptly modeled as a heterogeneous graph, in which we identify three types of vertices (i.e., news, entities, and topics), each associated with rich textual features, and three types of relations (i.e., news-news, news-entity, news-topic). Regarding the critical facts that fake news fabricates irregular relations among loosely related entities [12], and their semantics are deviating from the genuine, in this work, we are interested in such atypical local structures and reformulate fake news detection as classifying the heterogeneous subgraphs

---

[1]https://reutersinstitute.politics.ox.ac.uk/digital-news-report/2023
[2]https://www.theguardian.com/technology/2020/apr/06/at-least-20-uk-phone-masts-vandalised-over-false-5g-coronavirus-claims
[3]For clarity, we use news, news articles and news node interchangeably to denote a specific news piece to be classified in the rest of the paper

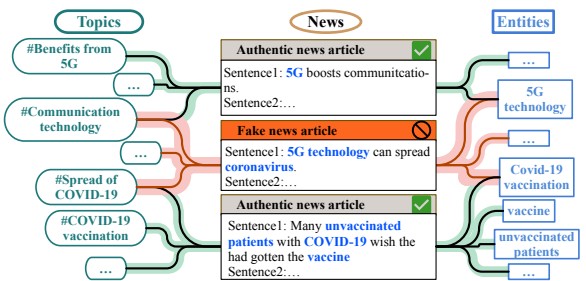

**Figure 1: Fake news forms atypical subgraph among seldom related news, entities, and topics. The fake news links the topic '#Spread of COVID-19' with entity '5G technology' in this case.**

centered on each news. As a concrete example depicted in Fig. 1, the subgraph rooted in the fake news article comprises the rarely related entities '5G' and 'COVID-19' and the topic '#Spread of COVID-19'. However, identifying and matching these subgraphs rely on the investigation of the heterogeneous graph, which is NP-hard.

In this work, we propose a novel **Hetero**geneous **S**ub**G**raph **T**ransformer, **HeteroSGT** for short, to overcome the prior challenges in textual feature learning and heterogeneous subgraph classification, particularly for the purpose of fake news detection. We first extract news entities and topics from all news articles and construct a heterogeneous graph for the subsequent subgraph mining. At this stage, we also employ a pre-trained dual-attention language model and digest the word-level and sentence-level semantics in each news article to obtain effective news features. We then apply random walk with restart (RWR) to extract subgraphs centered on each news, which explores the local graph structure in width and depth simultaneously. It is worth noting that through RWR, we can promptly acquire the relative positions of each node in the subgraph upon their positions in the RWR sequence. Thus, HeteroSGT introduces no further cost for obtaining the positional encoding compared to other Transformers for graph learning purposes [16, 20, 25]. The extracted subgraphs, represented as RWR sequences, are further taken as input to train our proposed subgraph Transformer together with partially observed labels. In a nutshell, our major contributions are:

- To the best of our knowledge, HeteroSGT is the first attempt to explore both the word- and sentence-level semantic patterns as well as the structural information among news, entities and topics for fake news detection. By modeling such information as a heterogeneous graph, HeteroSGT offers an effective solution to detect fake news through the investigation of the irregular subgraph structure and features.
- By assigning a relative positional encoding to each node with regard to its position in an RWR sequence, HeteroSGT mitigates the problem of learning node positional encodings in graph Transformers. Such a positional encoding reflects the relative closeness of a node to the target news in a particular subgraph, while maintaining the diversity in different RWR sequences. No cost for manifesting this encoding further advances our method to the state of the arts.
- Through extensive experiments on five mostly-used real-world datasets, we demonstrate the superior performance

of HeteroSGT over five baselines with regard to accuracy, macro-precision, macro-recall, macro-F1, and ROC. Further analysis and case studies validate the overall design choices of HeteroSGT, supported by the ablation study on its key components.

## 2 RELATED WORK

### 2.1 Content-based Fake News Detection

Content-based methods transform fake news detection into a text classification task, primarily leveraging the textual content of news articles. These methods typically employ natural language processing (NLP) techniques to extract diverse features, including linguistic, syntactic, stylistic, and other textual cues, from the content of news articles. For instance, Kim [15] proposed a model based on convolutional neural networks(CNNs) to capture local linguistic features from input text data. Horne *et al.* [10] conducted an analysis of differences in word usage and writing style between fake and real news, upon which fake news with distinguishable language styles can be detected. Kaliyar *et al.* [13] combined BERT [7] with three parallel CNN blocks to explore the news content. Yang *et al.* [38] built a dual-attention model and achieved improved performance by extracting hierarchical features of textual content. Along this line of research, others also investigate the integration of auxiliary textual information associated with news articles, such as comments [27, 28], and emotion signals [40], to further enhance the detection.

These content-based methods strive to explore diverse textual features associated with each single article to identify their authenticity. However, in the context where fake news is specially fabricated to mimic the semantics and language styles of genuine news, the detection performance is unsatisfying. Such defective results typically come from the unexplored relations between news articles and related entities in the news, such as entities and topics.

### 2.2 Graph-based Fake News Detection

Apart from the news article and its related content, we categorize other methods that explore the potential structures, such as word-word relations, news dissemination process, and social structure, as graph-based fake news detection. Representative works in exploring the word-word relations include Yao *et al.* [39], in which they first construct a weighted graph using the words contained within the news content and then apply the graph convolutional network (GCN) for classifying fake news. Hu *et al.* [18] built a similar graph but employed a heterogeneous graph attention network for classification. These methods consider the structure of the parse tree of each news article but still neglect the relations among different news.

Besides, Ma *et al.* [19] and Bian *et al.* [2] respectively employ recursive neural networks and bi-directional GCN to capture the new features in terms of their propagation process. Although the propagation graph of news characterizes the divergent features of fake news, these methods need to keep monitoring the dissemination of each news piece, which inherently limits their empirical applications. Other works also model the relations between news and users [8, 29, 30], or even news and external knowledge [9, 11, 33, 35, 36]

for fake news detection. Despite their advancement, their dependency on additional sources remains a significant obstacle.

## 3 METHODOLOGY

In this section, we first provide the preliminaries of this work, followed by details of our method. The overall framework of Het-eroSGT is depicted in Fig. 2. We first extract news, topics and entities from the articles (ⓐ, in Section 3.2) and derive the textual features of news articles considering both word-level and sentence-level semantics (ⓑ, Section 3.3). Then, we adopt RWR to extract subgraphs centered on each news (ⓒ, Section 3.4), and propose a novel subgraph transformer to generate effective latent representations for these subgraphs (ⓓ, Section 3.5). Eventually, fake news is distinguished as those with atypical subgraph patterns (Section 3.6).

### 3.1 Preliminaries

*3.1.1 News heterogeneous graph.* News articles naturally encompass various real-world **entities**, such as people, locations, or organizations. Furthermore, each piece of news typically involves specific domains or **topics**. These named entities and topics comprise rich macro-level semantic information and knowledge about news articles.

Considering the presence of entities and topics in each news article and the textual similarity among articles. In order to utilize such information for fake news detection, we propose to construct a heterogeneous graph for a given dataset, referring as $\mathcal{HG} = \{\mathbb{V}, \mathbb{L}\}$, to model the complex relations between news articles, entities and topics and the affluent features associated with them. $\mathbb{V} = \{\{n_i\}_{i=0}^{|\mathbb{N}|}, \{e_i\}_{i=0}^{|\mathbb{E}|}, \{t_i\}_{i=0}^{|\mathbb{T}|}\}$ is the node set, in which $\mathbb{N}$, $\mathbb{E}$ and $\mathbb{T}$ are the sets of news articles, entities, and topics, respectively. $n_i$, $e_i$, and $t_i$ denote a particular news, entity, and topic node in each set. $\mathbb{L}$ denotes the edge set containing direct links between the nodes, in which the edge types are *news-news, news-entity,* and *news-topic.*

*3.1.2 Fake news detection.* In this work, we define fake news detection as learning a classification function $f : \mathbb{V} \times \mathbb{L} \rightarrow \boldsymbol{y}$ using a set of labeled training news, i.e., $\mathbb{N}_{train} = \{n_i, y_i\}_{i=0}^{|\mathbb{N}_{train}|}$. $y_i \in \{0, 1\}$ is the label of a corresponding news article $n_i$. Label 1 denotes a fake news article, and 0 for authentic news. After sufficient training, $f$ interpolates the labels of other news in the dataset.

### 3.2 News Heterogeneous Graph Construction

Our constructed heterogeneous graph comprises the following three types of nodes and relations.

**News nodes**. In $\mathcal{HG}$, we represent each news article as a news node $n_i$. Given $m$ pieces of news, it can be presented as $\mathbb{N} = \{n_1, n_2, \cdots, n_m\}$ ($\mathbb{N} \subset \mathbb{V}$). In particular, we construct an edge (*news-news*) between two news articles if their shared number of entities exceeds the average level or the topics they focus on are the same.

**Entity nodes**. We use the SpaCy library [4] to extract all entities from the news articles, each of which is denoted as an entity node $e_i \in \mathbb{E}$. If entity $e_i$ is mentioned in the news article $n_i$, we build a *news-entity* edge between them.

**Topic nodes**. We employ the Latent Dirichlet Allocation (LDA) model [3] to derive potential topics involved in all news by setting

[4] https://spacy.io

the number of total topics set as $k$. Empirically, the best $k$ can be acquired through the analysis of topic coherence and perplexity, and our choices are reported in Section 4.5. We denote each topic as a topic node $t_i \in \mathbb{T}$ and build a *news-topic* edge if the topic belongs to the top $\lambda_t$ most related topics of a particular news based on the output of the LDA model.

We further obtain the features of these three types of nodes for the subsequent subgraph learning. Specifically, we present a pre-trained dual-attention module to learn features of news nodes (in section 3.3) and initialize the entities and topics' features using the BERT model [7].

### 3.3 Dual-attention News Embedding Module

We employ the news embedding module in [38] to get the features of each news article progressively regarding the attentions of words to the sentence (word-level) and sentence to the article (sentence-level).

*3.3.1 Word-level attention.* Given news $n_i$ that consists of $s_i$ sentences, the $p$-th sentence of $n_i$ is denoted as $s_p = [w_{p,1}, w_{p,2}, \ldots, w_{p,q}]$, where $w_{p,q}$ is the $q$-th word in $p$-th sentence of $n_i$. We initialize an embedding matrix $\boldsymbol{H}_w$ for the words and apply BiGRU [6] to update them by

$$\boldsymbol{h}_{w_{p,q}} = [\overrightarrow{\boldsymbol{h}}_{w_{p,q}}, \overleftarrow{\boldsymbol{h}}_{w_{p,q}}] = BiGRU(\boldsymbol{H}_w), \quad (1)$$

where $\boldsymbol{h}_{w_{p,q}}$ is the learned word vector of $w_{p,q}$. $\overrightarrow{\boldsymbol{h}}_{w_{p,q}}$, and $\overleftarrow{\boldsymbol{h}}_{w_{p,q}}$ are the outputs of BiGRU in the forward and reverse direction. As words in one sentence do not contribute equally to the entire meaning of the sentence, we then introduce the word-level attention mechanism to obtain the sentence embedding $\boldsymbol{h}_{s_p}$ following:

$$\boldsymbol{u}_{w_{p,q}} = tanh(\boldsymbol{h}_{w_{p,q}}\boldsymbol{W}_w + \boldsymbol{b}_w), \quad (2)$$

$$\alpha_{w_{p,q}} = \frac{exp(\boldsymbol{u}_{w_{p,q}}^\top \boldsymbol{u}_w)}{\sum_{j=1}^q exp(\boldsymbol{u}_{w_{p,j}}^\top \boldsymbol{u}_w)}, \quad \boldsymbol{h}_{s_p} = \sum_{j=1}^q \alpha_{w_{p,j}} \boldsymbol{h}_{w_{p,j}}, \quad (3)$$

where $\alpha_{w_{p,q}}$ denotes normalized importance of word $w_{p,q}$ to the sentence $s_p$. $\boldsymbol{W}_w$ and $\boldsymbol{b}_w$ are learnable parameters. $\boldsymbol{u}_w$ denotes the randomly initialized word context vector, which is learned jointly during training. Consequently, we obtain the sentence vector $\boldsymbol{h}_{s_p} \in \boldsymbol{H}_s$ as a weighted sum of the word vectors within it.

*3.3.2 Sentence-level attention.* Considering the divergent importance of sentences to the whole news article, we further apply a sentence-level attention mechanism to extract features for the whole article. Similar to the word-level BiGRU, we apply another BiGRU to encode the sentence order information in the article by

$$\boldsymbol{z}_{s_p} = [\overrightarrow{\boldsymbol{z}}_{s_p}, \overleftarrow{\boldsymbol{z}}_{s_p}] = BiGRU(\boldsymbol{H}_s), \quad (4)$$

and then measure the sentence-level attention following:

$$\boldsymbol{u}_{s_p} = tanh(\boldsymbol{z}_{s_p}\boldsymbol{W}_s + \boldsymbol{b}_s), \quad (5)$$

$$\alpha_{s_p} = \frac{exp(\boldsymbol{u}_{s_p}^\top \boldsymbol{u}_s)}{\sum_{j=1}^{|n|} exp(\boldsymbol{u}_{s_j}^\top \boldsymbol{u}_s)}, \quad \boldsymbol{h}_{n_i} = \sum_{j=1}^{|n|} \alpha_{s_p} \boldsymbol{z}_{s_p}, \quad (6)$$

where $\boldsymbol{h}_{n_i}$ is the learned features of news $n_i$ and $|n|$ denotes the number of sentences in it. $\boldsymbol{u}_s$ is the sentence context vector and similar to that for word context, it is randomly initialized and learned throughout training. All the parameters in the BiGRUs, $\{\boldsymbol{W}\}_{w,s}$ and

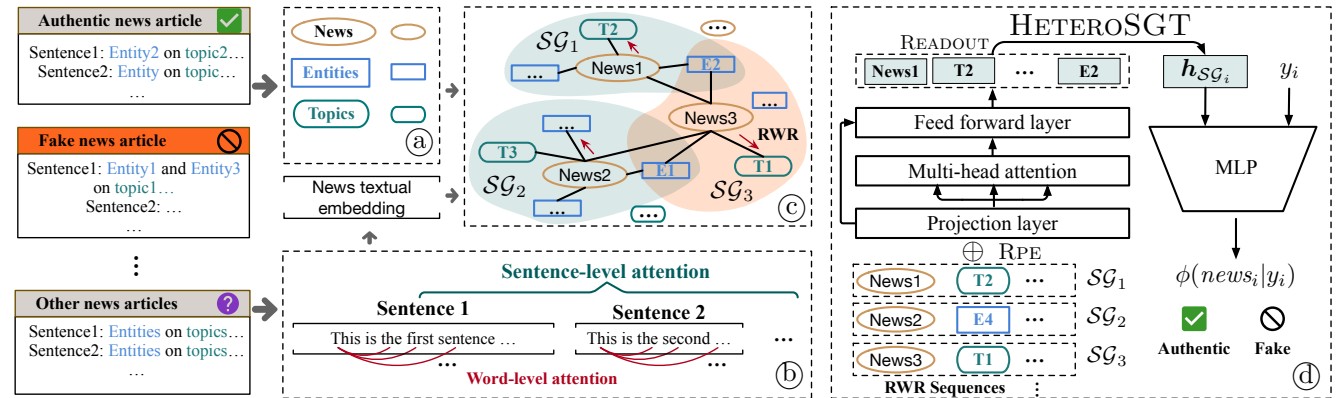

**Figure 2: Overall framework of HETEROSGT. ⓐ News, entities, and topics are extracted from all new articles. ⓑ The pre-trained dual-attention module derives news embeddings considering both word-level and sentence-level semantics. ⓒ A heterogeneous graph $\mathcal{HG}$ is constructed to model the relations among news, entities, and topics, after which we initiate RWR (→) centered on each news to extract subgraphs. ⓓ HETEROSGT takes the RWR sequences as input and generates a subgraph representation $h_{\mathcal{SG}_i}$ to train the MLP classifier with observed labels for detecting fake news.**

$\{b\}_{w,s}$ are fine-tuned to minimize the cross-entropy on the training set, following Eq. 15.

## 3.4 RWR-based Heterogeneous Subgraph Sampling

Given our constructed heterogeneous graph $\mathcal{HG}$ and recall that fake news usually form atypical subgraphs that rarely contain co-related topics and entities. Since subgraph matching is NP-hard, especially on heterogeneous graphs, we first propose an RWR-based sampling algorithm to extract heterogeneous subgraphs centered on each news article and then compare these subgraphs with regard to their latent representations for fake news detection.

In RWR, we initiate the root of each walk as a news node and uniformly sample neighboring nodes on the $\mathcal{HG}$, i.e., topics, entities, and other news, to join the walk until the total walk length reaches $wl$. For an effective exploitation of both the width and depth of the subgraph centered on each news, we set a restart probability of $r$. Denoting the $i$-th node in the walk as $v_i$, the possibility of the next node to join the RWR sequence is given by

$$p(v_j|v_i) = \begin{cases} 1-r, & with \quad (v_j, v_i) \in \mathbb{L} \\ r, & v_j = v_{root} \end{cases} \tag{7}$$

where $(v_j, v_i)$ denotes the link between $v_i$ and $v_j$, $v_{root}$ is the news node this RWR centered on. In doing so, subgraph $\mathcal{SG}_i$ can be firmly presented as a matrix $S_i$, in which each row $S_i^j$ corresponds to the feature of the $j$-th node in the RWR sequence. This algorithm is summarized in Alg. 1 and we further validate the impact of the restart probability $r$ in Section 4.6

## 3.5 Heterogeneous Subgraph Transformer

Each of our extracted subgraphs $\mathcal{SG}_i$ contains affluent information about the news article and its related topics, entities, and other news. To identify the atypical subgraphs centered on fake news, one may consider applying neural networks to encode the spatial structure and attributes of subgraphs into latent representations and expect

---

**Algorithm 1:** RWR-based Heterogeneous Subgraph Sampling

**Input:** $\mathcal{HG}$: news heterogeneous graph; $wl$: the length of random walk; news set $\mathbb{N}$.

**Output:** $\mathcal{SG}_i \in \mathbb{SG}$: sampled subgraph centered on each news article $n_i$ and the subgraph set.

1 **for** $n_i \in \mathbb{N}$ **do**
2 $\quad v_{root} \leftarrow n_i$.
3 $\quad$ **while** $|\mathcal{SG}_i| \leq wl$ **do**
4 $\quad\quad \mathcal{SG}_i \leftarrow$ Sample next node $v_j$ by Eq. 7
5 $\quad$ **end**
6 $\quad \mathbb{SG} \leftarrow \mathbb{SG} \cup \mathcal{SG}_i$
7 **end**
8 **return** $\mathbb{SG}$

---

subgraphs corresponding to fake news located in deviating locations compared the authentic news in the latent space. However, the design choices of the neural network architecture are highly constrained by the heterogeneity of nodes and their complex relations in $\mathcal{SG}_i$. To be specific, the designed neural networks should be capable of 1) exploiting the heterogeneous links and features associated with each type of nodes; and 2) adaptively summarizing such heterogeneous information to interpolate the subgraph representations. In this work, we propose a novel heterogeneous subgraph transformer, HETEROSGT, to fulfill these objectives. It has the following three key ingredients.

*3.5.1 Relative positional encoding on RWR.* The position of each node in an extracted RWR sequence unveils its relation with the centering news article. Taking the second node in the RWR sequence as a concrete example, its direct link with the target news illustrates that this node, which might be a particular topic, entity, or another news article, has a closer relation with the target news than other nodes of the same type in the present sequence. We, therefore, directly utilize such a relative position to derive their

positional encoding in the subgraph. It is noteworthy that the positional encoding of each node is still an open problem for graph transformers [16, 20, 25]. Rather, our positional encoding method is aware of the local subgraph patterns concerning the target news article, and the positional encodings of the same node in different sequences are also divergent because of their varying presence. The efficacy of this positional encoding method is further validated in Section 4.7.

Practically, we follow the sinusoidal encoding in Transformer [31] to get the relative positional encoding by

$$\text{RPE}_{j,2i} = \sin(l/10000^{2/d}) \quad (8)$$

$$\text{RPE}_{j,2i+1} = \cos(l/10000^{2/d}), \quad (9)$$

where $j \in \{1, \ldots, wl\}$ denotes the position of the node in the RWR sequence, $i$ is the dimension in vector $\text{RPE}_j$, and $d$ is the dimensionality of node features.

*3.5.2 Heterogeneous self-attention module.* For the heterogeneity of nodes in $\mathcal{SG}_i$, we first propose to project them into the same space and learn their latent representations with regard to their mutual attention. In particular, we take the advantage of multi-head self-attention module in Transformer and employ the keys, queries, and values' projections, i.e., $Q, K, V$, for eliminating the heterogeneity of node features at the same time as updating each node's representation following:

$$\text{ATTN}(H_i^{l-1}) = \text{Softmax}(\frac{Q_i K_i^\top}{\sqrt{d}}), H_i^l = \text{FFN}[\text{ATTN}(H_i^{l-1})V_i] \quad (10)$$

$$\text{with} \quad Q_i = H_i^{l-1} W_Q, K_i = H_i^{l-1} W_K, V_i = H_i^{l-1} W_V, \quad (11)$$

where $W_Q, W_K, W_V \in \mathbb{R}^{|\mathcal{SG}_i| \times d}$ are the projection matrices corresponding to keys, queries, and values, respectively. $\text{FFN}(\cdot)$ is the feed forward neural network with residual links and layer normalization following the conventional Transformer architecture. $H_i^l$ is the output of the $l$-th attention block, in which each row contains the transformed representations of each node. $H_i^0 = S_i \oplus \text{RPE}$ is the feature matrix after adding the relative positional encoding.

To unleash the power of multi-head attention for downgrading the randomness of the initialized attention weights, we can promptly extend this module to work on multi-heads by

$$H_i^l = \Big\|_{h=1}^{Heads} H_i^{l,h}, \quad (12)$$

where $H_i^{l,h}$ is the representations learned using head $h$, and $\big\|$ denotes the concatenation operation.

*3.5.3 Adaptive subgraph representation generation.* Given the node representations learned within each subgraph, our primary goal is to read out a vector representation for the whole subgraph from the last Transformer layer $L$, which is given by

$$h_{\mathcal{SG}_i} = \text{READOUT}(H_i^L). \quad (13)$$

As mentioned above, this essential READOUT function should take account of the different contributions of different types of nodes and relations. By delving deeper into the heterogeneous self-attention module, we can see that the representation of each node already encapsulates the relations and features associated with news articles, topics, and entities within the whole subgraph,

---

**Algorithm 2:** The training process of HETEROSGT

**Input:** $\mathbb{SG}$: Subgraphs extracted from $\mathcal{HG}$; $S_i$: Feature matrix of nodes in subgraph $\mathcal{SG}_i$; $y$: labels of training news.

**Output:** Trained classifier $f$.

1 **do**
2    **for** $\mathcal{SG}_i \in \mathbb{SG}$ **do**
3      RPE ← relative positional encoding by Eqs. 8 and 9
4      $H_i^0 \leftarrow S_i \oplus \text{RPE}$ // *Add positional encoding*
5      $Q_i, K_i, V_i \leftarrow$ Projected Query, Key, and Value matrices by Eq. 11
6      **for** *Transformer layers* $l \in \{1, \ldots, L\}$ **do**
7        $H_i^l \leftarrow$ Updated node representations by Eq. 10
8      **end**
9      $h_{\mathcal{SG}_i} \leftarrow \text{READOUT}(H_i^L)$
10      $\phi(n_i|y_i) \leftarrow f_{\text{MLP}}(h_{\mathcal{SG}_i})$
11      Take gradient step based on Eq. 15
12    **end**
13 **until** *converged*;

---

benefiting from the global attention mechanism of the Transformer backbone. Moreover, since all our random walks are initiated at news articles, the first row of $H_i^L$ constantly represents the representation of the news article this subgraph $\mathcal{SG}_i$ is centered on. Both observations imply that we can directly take the representation of the target news article as the subgraph representation for further detection. We further compare it with the mostly-used mean and max readout functions and show that such a simple method is effective without incurring any additional computational overhead (see Section 4.8).

## 3.6 Fake News Detection

After deriving the subgraph representation, we train a two-layer MLP to predict the authenticity of each news article following:

$$\phi(n_i|y_i) = f_{\text{MLP}}(h_{\mathcal{SG}_i}) = \text{Softmax}[(h_{\mathcal{SG}_i} W_1 + b_1)W_2 + b_2], \quad (14)$$

where $W_1, W_2, b_1$ and $b_2$ are learnable parameters in the MLP, which are fine-tuned together with the parameters in our subgraph Transformer (i.e., $W_Q, W_K, W_V$) to minimize the cross-entropy loss:

$$\mathcal{L} = -\sum_{y \in \{0,1\}} \sum_{n_i \in \mathbb{N}_{train}} \frac{1}{|\mathbb{N}_{train}|} \log \phi(n_i \mid y_i), \quad (15)$$

where $\mathbb{N}_{train}$ is the trainning set. For brevity, we summarize the algorithm of HETEROSGT in Alg. 2.

## 4 EXPERIMENT

We conduct extensive experiments on five real-world datasets to demonstrate the efficacy of HETEROSGT. We first outline the experimental setup, including the datasets and baselines. Subsequently, we report the overall results, followed by four specific case studies that elucidate our design choices. The ablation study further delineates the ingredients contributing to the performance improvement and the parameter analysis validates our model's sensitivity to the key parameters.

**Table 1: Statistics of Datasets.**

| Dataset | # Fake | # Real | # Total | # Topics ($k$) |
|---------|--------|--------|---------|----------------|
| MM COVID | 1,888 | 1,162 | 3,048 | 50 |
| ReCOVery | 605 | 1,294 | 1,899 | 35 |
| MC Fake | 2,671 | 12,621 | 15,292 | 40 |
| PAN2020 | 250 | 250 | 500 | 25 |
| LIAR | 2,507 | 2,053 | 4,560 | 50 |

## 4.1 Dataset

The five datasets encompass a wide range of subject areas, including health-related datasets (MM COVID [17] and ReCOVery [41]), a political dataset (LIAR [32]), and multi-domain datasets (MC Fake [22] and PAN2020 [26]). The details statistics of these datasets are summarized in Table 1. It is worth noting that the MC Fake dataset comprises news articles covering politics, entertainment, and health, which are sourced from well-known debunking websites such as PolitiFact[5] and GossipCop [22].

## 4.2 Baselines

For a fair evaluation of the overall detection performance and considering the availability of additional sources, we compared HETEROSGT with four baselines that detect fake news using only news text and HGNNR4FD, which involves open-sourced knowledge graphs for fake news detection. The baselines include:

- **textCNN** [15] employs multiple convolutional neural network layers (CNNs) to extract news features from sentences, which are then applied for news classification.
- **textGCN** [39] constructs a weighted graph based on news articles and the words within them, in which the edge weights are measured by TF-IDF (term frequency–inverse document frequency) values. Subsequently, it utilizes a graph convolutional network for text classification.
- **HAN** [38] explores the word-sentence and sentence-article structures in news articles through BiGRUs. It considers the contributions of different words to sentence-level embeddings and the contributions of sentences to learn the embedding of the whole article. Fake news and genuine news are distinguished with respect to their embeddings.
- **Bert** [7] is the well-known transformer-based language model. In our experiment, we use the [CLS] token's embedding for classifying fake news.
- **HGNNR4FD** [35] represents news content as a heterogeneous graph and utilizes external entity knowledge from Wikidata5M to learn news representations.

It is worth mentioning that we do not include other baselines that rely on the propagation information [34, 37], social engagement [28, 40], and other sources of evidence [14, 36] for a fair comparison. We also ignore other conventional heterogeneous graph neural networks, such as heterogeneous graph attention neural networks, because HGNNR4FD has already demonstrated superiority over them.

---

[5]https://www.politifact.com/

## 4.3 Experimental Settings

*Model Configuration.* For constructing the news heterogeneous graph, we set the optimal number of topics, denoted as $k$, for each dataset through topic model evaluation (see Table 1). We set $\lambda_t$, which decides whether a news node connects to a topic node, as 3. The news embedding size of the dual-attention news embedding module is set to 600, and we employ the Adam optimizer for training, with a learning rate of 5e-5 and weight decay of 5e-3. Unless otherwise stated, we use the same parameter settings across all baselines to ensure a fair comparisons.

All of the datasets are divided into train, validation, and test sets using a split ratio of 80%-10%-10%. To test the generalizability, we perform 10 rounds of tests with random seeds on each model and report the averaged results and standard deviation. All the experiments are conducted on 1 NVIDIA Volta GPU with 30G RAM.

*Evaluation Metrics.* We evaluate each model's performance with regard to accuracy (Acc), macro-precision (M-Pre), macro-recall (M-Rec), macro-F1 (M-F1), and AUC-ROC curve.

## 4.4 Overall Results

Table 2 presents the results of our proposed model HETEROSGT and the baselines across the five datasets. We can see that HETEROSGT attains the best performance across all metrics on all datasets, except for the second-best M-Rec on the PAN2020. This affirms that the structural information and textual features derived from news-centered subgraphs are indeed valuable for discerning the authenticity of news articles, which contribute to the outstanding performance of HETEROSGT. It should be noted that HETEROSGT consistently achieved a remarkably higher level of recall compared to other models across most cases, particularly on the MC Fake dataset. A high recall is usually favored when combating the dissemination of fake news because it ensures that fewer fake news are overlooked. In addition, while some baselines experience fluctuations in their results, HETEROSGT is able to produce dependable and consistent results.

As for the baseline models, TextCNN has relatively poor performance across multiple datasets. This may stem from its reliance on fixed-length convolutional kernels for feature extraction, potentially limiting its capacity to capture the long-range dependencies and grasp the holistic textual context. TextGCN exhibits varying performance outcomes, delivering strong results on MC Fake and showing notable discrepancies in recall and precision on other datasets. Both HAN and Bert utilize attention mechanisms, and their overall performance is comparable. HGNNR4FD achieves suboptimal results on most datasets due to its incorporation of external knowledge to enhance news representations. However, when comparing its performance with HETEROSGT across five datasets, it exhibits an average decrease of 5.4% on accuracy, 8.6% on macro-precision, 6.1% on macro-recall, and 7.6% on macro-F1.

## 4.5 Case Study I - Topic Model Evaluation

In this case study, we adopt a dual-metric approach, incorporating both perplexity and coherence, to evaluate the topic modeling process in HETEROSGT. Perplexity measures the model's predictive performance, while coherence assesses the interpretability of the generated topics. The evaluation was conducted over a range of topic

**Table 2: Detection Performance on Five Datasets (Best in Red, Second-best in Blue).**

| Dataset | TextCNN | | TextGCN | | BERT | | HAN | | HGNNR4FD | | HeteroSGT | |
|---|---|---|---|---|---|---|---|---|---|---|---|---|
| | Acc | M-Pre | Acc | M-Pre | Acc | M-Pre | Acc | M-Pre | Acc | M-Pre | Acc | M-Pre |
| **MM COVID** | 0.582±0.035 | 0.478±0.170 | 0.717±0.156 | 0.735±0.236 | 0.730±0.093 | 0.727±0.094 | 0.855±0.005 | 0.854±0.005 | 0.722±0.016 | 0.894±0.043 | **0.925±0.004** | **0.921±0.006** |
| **ReCOVery** | 0.658±0.011 | 0.460±0.104 | 0.718±0.037 | 0.691±0.178 | 0.682±0.030 | 0.441±0.213 | 0.722±0.021 | 0.462±0.197 | 0.795±0.019 | 0.789±0.030 | **0.909±0.002** | **0.902±0.002** |
| **MC Fake** | 0.825±0.001 | 0.544±0.156 | 0.724±0.138 | 0.552±0.169 | 0.827±0.006 | 0.713±0.271 | 0.825±0.005 | 0.463±0.098 | 0.824±0.008 | 0.412±0.004 | **0.883±0.002** | **0.812±0.003** |
| **PAN2020** | 0.514±0.022 | 0.3123±0.116 | 0.551±0.013 | 0.318±0.391 | 0.510±0.044 | 0.523±0.049 | 0.508±0.046 | 0.473±0.132 | 0.671±0.025 | 0.681±0.037 | **0.728±0.010** | **0.737±0.007** |
| **LIAR** | 0.546±0.019 | 0.432±0.181 | 0.550±0.068 | 0.147±0.243 | 0.537±0.007 | 0.513±0.017 | 0.546±0.025 | 0.493±0.036 | 0.575±0.013 | 0.546±0.022 | **0.581±0.002** | **0.580±0.003** |
| Dataset | M-Rec | M-F1 | M-Rec | M-F1 | M-Rec | M-F1 | M-Rec | M-F1 | M-Rec | M-F1 | M-Rec | M-F1 |
| **MM COVID** | 0.547±0.039 | 0.474±0.101 | 0.685±0.178 | 0.622±0.241 | 0.722±0.101 | 0.720±0.103 | 0.854±0.006 | 0.853±0.005 | 0.632±0.040 | 0.739±0.0175 | **0.915±0.005** | **0.918±0.005** |
| **ReCOVery** | 0.501±0.020 | 0.442±0.039 | 0.609±0.102 | 0.565±0.124 | 0.506±0.012 | 0.416±0.032 | 0.501±0.007 | 0.425±0.011 | 0.742±0.051 | 0.747±0.047 | **0.886±0.005** | **0.893±0.003** |
| **MC Fake** | 0.501±0.002 | 0.455±0.004 | 0.516±0.020 | 0.470±0.039 | 0.501±0.001 | 0.454±0.001 | 0.500±0.001 | 0.453±0.001 | 0.500±0.000 | 0.452±0.002 | **0.762±0.002** | **0.783±0.003** |
| **PAN2020** | 0.502±0.007 | 0.3597±0.042 | 0.263±0.032 | 0.286±0.035 | 0.515±0.044 | 0.491±0.040 | 0.517±0.031 | 0.460±0.086 | **0.781±0.021** | 0.727±0.036 | 0.736±0.010 | 0.728±0.010 |
| **LIAR** | 0.502±0.005 | 0.377±0.049 | 0.212±0.418 | 0.138±0.247 | 0.510±0.012 | 0.483±0.014 | 0.502±0.018 | 0.445±0.053 | 0.507±0.022 | 0.526±0.009 | **0.575±0.002** | **0.571±0.003** |

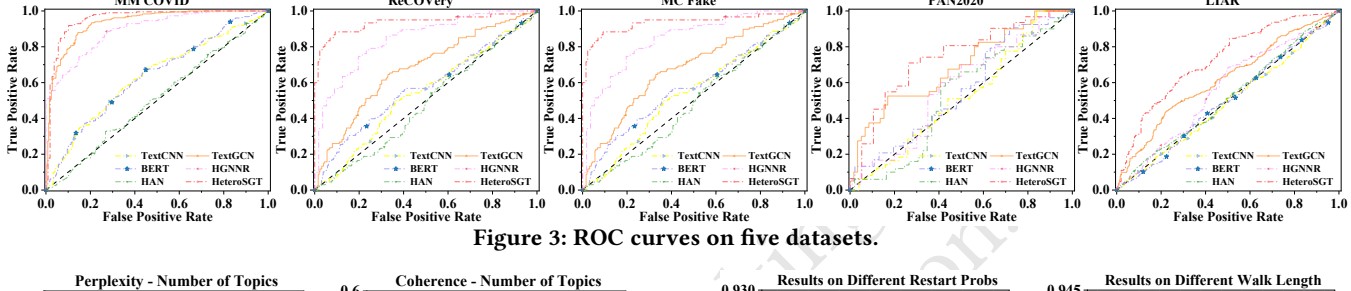

**Figure 3: ROC curves on five datasets.**

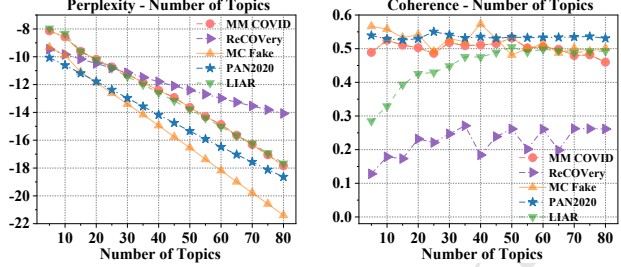

**Figure 4: Topic model evaluation.**

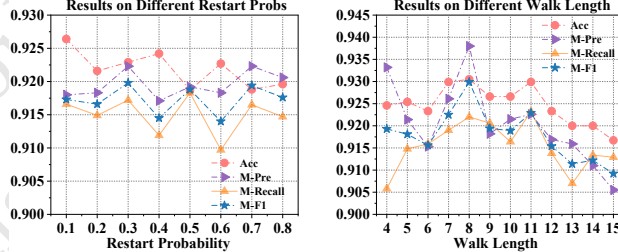

**Figure 5: Impact of RWR length and restart probability.**

numbers, spanning from 5 to 80, with increments of 5. Typically, the optimal number of topics corresponds to the point at which perplexity decreases most rapidly or where coherence achieves its highest value. As shown in Fig. 4, perplexity consistently decreases with the growing number of topics across all datasets, while the coherence scores peak at different topic numbers on different datasets. To balance the model performance and topic interpretability, our selection of the optimal topic number for each dataset was based on the point at which coherence scores reach their peak.

## 4.6 Case Study II - Impact of RWR walk length and restart probability

The random walk length and restart probability regulate the subgraphs extracted from $\mathcal{HG}$. Specifically, the walk length determines the size of each subgraph by specifying the number of neighboring nodes to be included. On the other hand, the restart probability plays a crucial role in determining the search direction, i.e., whether the exploration should focus on local or long-distance information. To evaluate their impact on the detection performance, we conducted a case study on MM COVID by varying the walk length from 4 to 15 and the restart probability from 0.1 to 0.8. From the

results (on the validation set) depicted in Fig. 5, we see that the walk length and restart probability merely affect the detection results, which vary between 0.905 and 0.940 on the four evaluation metrics. For MM COVID, the optimal walk length is 11, and the restart probability is 0.1, which suggests involving 11 nodes in each subgraph and prior the exploration of $\mathcal{HG}$ in depth to attain better performance.

## 4.7 Case Study III - Efficacy of our relative positional encoding

The order of each node in the RWR sequence reflects its relevance to the centering news node in the extracted subgraph. Similar to the positional encoding in the conventional Transformer, our proposed relative positional encoding is to capture such relevance. This study targets on evaluating the efficacy of our proposed relative positional encoding module by comparison between HETEROSGT and HETEROSGT⊘RPE, which neglects positional encoding for learning. From the results in Table 3, we see that the detection performance with RPE is constantly better. Regarding the firmly 1% improvement, as an open problem in graph Transformer [16, 20, 25], we leave more effective positional encoding methods to future works.

**Table 3: Results of case studies III and IV on MM COVID.**

| Methods | Acc | M-Pre | M-Rec | M-F1 |
|---|---|---|---|---|
| HETEROSGT | **0.925±0.004** | **0.921±0.006** | **0.915±0.005** | **0.918±0.005** |
| HETEROSGT ⊘ RPE | 0.917±0.006 | 0.912±0.008 | 0.905±0.006 | 0.908±0.007 |
| HETEROSGT$_{mean}$ | 0.720±0.019 | 0.693±0.022 | 0.682±0.020 | 0.685±0.021 |
| HETEROSGT$_{max}$ | 0.732±0.019 | 0.708±0.022 | 0.686±0.022 | 0.700±0.022 |

**Table 4: Ablation Results**

| Datasets | Methods | Acc | M-Pre | M-Rec | M-F1 |
|---|---|---|---|---|---|
| MM COVID | HETEROSGT ⊘ $\mathcal{HG}$ | 0.867±0.084 | 0.841±0.139 | 0.843±0.123 | 0.836±0.141 |
| | HETEROSGT ⊘ E&T | 0.900±0.006 | 0.892±0.007 | 0.873±0.007 | 0.882±0.006 |
| | HETEROSGT ⊘ E | 0.918±0.006 | 0.918±0.008 | 0.901±0.007 | 0.909±0.007 |
| | HETEROSGT ⊘ T | 0.921±0.006 | 0.915±0.007 | 0.912±0.007 | 0.914±0.007 |
| | HETEROSGT | **0.925±0.004** | **0.921±0.006** | **0.915±0.005** | **0.918±0.005** |
| ReCOVery | HETEROSGT ⊘ $\mathcal{HG}$ | 0.817±0.107 | 0.781±0.209 | 0.738±0.147 | 0.716±0.208 |
| | HETEROSGT ⊘ E&T | 0.884±0.008 | 0.878±0.009 | 0.848±0.011 | 0.861±0.010 |
| | HETEROSGT ⊘ E | 0.900±0.009 | 0.892±0.013 | 0.873±0.012 | 0.882±0.011 |
| | HETEROSGT ⊘ T | 0.905±0.06 | 0.893±0.06 | 0.886±0.07 | 0.889±0.07 |
| | HETEROSGT | **0.909±0.002** | **0.902±0.002** | **0.886±0.005** | **0.893±0.003** |
| MC Fake | HETEROSGT ⊘ $\mathcal{HG}$ | 0.828±0.017 | 0.524±0.169 | 0.568±0.104 | 0.542±0.136 |
| | HETEROSGT ⊘ E&T | 0.860±0.004 | 0.781±0.008 | 0.683±0.010 | 0.714±0.009 |
| | HETEROSGT ⊘ E | 0.864±0.005 | 0.777±0.009 | 0.721±0.016 | 0.743±0.014 |
| | HETEROSGT ⊘ T | 0.878±0.003 | 0.803±0.009 | 0.752±0.007 | 0.773±0.006 |
| | HETEROSGT | **0.883±0.002** | **0.812±0.003** | **0.762±0.002** | **0.783±0.003** |
| PAN2020 | HETEROSGT ⊘ $\mathcal{HG}$ | 0.506±0.077 | 0.505±0.085 | 0.514±0.075 | 0.479±0.090 |
| | HETEROSGT ⊘ E&T | 0.600±0.040 | 0.604±0.046 | 0.603±0.042 | 0.599±0.042 |
| | HETEROSGT ⊘ E | 0.620±0.032 | 0.622±0.033 | 0.622±0.031 | 0.620±0.031 |
| | HETEROSGT ⊘ T | 0.680±0.024 | 0.701±0.025 | 0.686±0.024 | 0.675±0.024 |
| | HETEROSGT | **0.728±0.010** | **0.737±0.007** | **0.736±0.010** | **0.728±0.010** |
| LIAR | HETEROSGT ⊘ $\mathcal{HG}$ | 0.528±0.015 | 0.493±0.035 | 0.502±0.017 | 0.445±0.051 |
| | HETEROSGT ⊘ E&T | 0.561±0.005 | 0.558±0.005 | 0.555±0.004 | 0.552±0.004 |
| | HETEROSGT ⊘ E | 0.568±0.010 | 0.565±0.012 | 0.563±0.009 | 0.561±0.008 |
| | HETEROSGT ⊘ T | 0.579±0.008 | 0.577±0.008 | 0.572±0.008 | 0.568±0.008 |
| | HETEROSGT | **0.581±0.002** | **0.580±0.003** | **0.575±0.003** | **0.571±0.003** |

### 4.8 Case Study IV - Comparisons between readout functions

As stressed in Section 3.5, the readout function is essential to fuse a subgraph-level representation from all nodes within it. We implement two additional variants of HETEROSGT by replacing our readout function as the conventional mean and max pooling functions, namely HETEROSGT$_{mean}$ and HETEROSGT$_{max}$. From Table 3, we see that simply read out the subgraph-level representation as the centering news representation is more effective than mean and max pooling, and most importantly, this method introduces no additional cost.

### 4.9 Ablation Study

To assess the impact of different components within the news heterogeneous graph, namely news articles, news topics, and news entities, we perform a series of ablation experiments involving the removal of specific nodes from the graph. Notationally, '⊘$\mathcal{HG}$' denotes the variant that excludes the news heterogeneous graph and only uses the output of the dual-attention module $\boldsymbol{h}_{n_i}$ for detecting fake news. Similarly, '⊘T&E' signifies a variant that drop all topic nodes and entity nodes from the news heterogeneous graph for learning. Lastly, '⊘T' and '⊘E' correspond to variants that respectively remove topic nodes and entity nodes.

Based on the results presented in Table 4, it is evident that the performance is less than satisfactory when we attempt to learn news embeddings without utilizing the news heterogeneous graph,

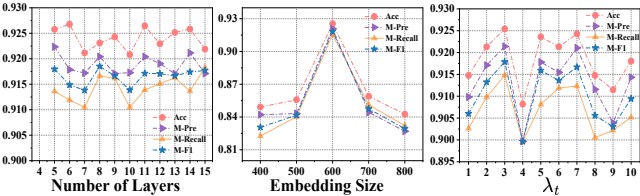

**Figure 6: HETEROSGT's sensitivity to three parameters.**

as in the case of HETEROSGT⊘$\mathcal{HG}$. Introducing the heterogeneous graph into the training process, as seen in HETEROSGT⊘E&T, leads to noticeable performance improvements across all the datasets. This improvement is consistent and becomes more significant as we incorporate additional features related to news topics and entities, as observed with HETEROSGT ⊘ E and HETEROSGT ⊘ T. It is noticeable that HETEROSGT ⊘ E, which retains topic nodes, consistently outperforms HETEROSGT ⊘ T which neglects entity nodes. Ultimately, the peak performance is achieved when all components are included in the HETEROSGT training process. In summary, all four components of HETEROSGT exhibit remarkable efficacy in the context of fake news detection. As we progressively introduce these components into the training process, we observe a steady and consistent improvement in the performance of fake news detection. This highlights the importance of leveraging the news heterogeneous graph, incorporating news topics and entities, and underscores the effectiveness of HETEROSGT as a comprehensive solution for fake news detection.

### 4.10 Parameter Sensitivity Analysis

Additionally, we conduct an investigation into the sensitivity of HETEROSGT concerning parameters: the number of transformer layers, the embedding size and the top $\lambda_t$. The findings indicate that the performance exhibits only minor fluctuations when varying the number of transformer layers. However, there are variations in performance when the embedding size of news changes. Regarding $\lambda_t$, we tested values from 1 to 10, and the results suggest that the model achieves its best performance when $\lambda_t$ is set to 3. Our approach consistently delivers satisfactory results when the news embedding size is 600 and the number of layers is 5.

## 5 CONCLUSION

In this work, we delved into the structural and textual features of news articles, utilizing a heterogeneous graph that connects news, topics and entities. We propose a heterogeneous subgraph transformer (HETEROSGT) to exploit the informative subgraphs for effective fake news detection. HETEROSGT integrates a pre-trained dual-attention news embedding module, enabling the extraction of textual features from both word-level and sentence-level semantics present in news articles. By incorporating random walks, our subgraph Transformer adeptly captures information from subgraphs surrounding each news article, enabling the detection of fake news in the subgraph representation space. Extensive experiments on five real-world datasets demonstrate the superiority of our method over baselines and the comprehensive case studies on the key components validate the design choices of HETEROSGT.

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
