# OpenReview forum: "Heterogeneous Subgraph Transformer for Fake News Detection"
_ACM.org/TheWebConf/2024/Conference — TheWebConf24 Oral_

### Official Review · Reviewer_89hq · 2023-11-22

**Novelty:** 6
**Technical Quality:** 6

**Review:**

The paper proposes the detection of fake news by identifying irregular subgraph structures and features in a heterogeneous graph. The graph captures word- and sentence-level semantic patterns and structural information among news, entities, and topics. The model is evaluated across 5 datasets that span various subject areas. The authors also perform 4 case studies, an ablation study, and a hyperparameter sensitivity analysis to better understand the strengths and limitations of their model.

The paper is written very well overall with very detailed experiments. Although the individual parts themselves are not necessarily new approaches (i.e., random walk subgraph sampling, heterogeneous graph transformer) there are two interesting tweaks that improve the model performance. The first is to leverage the random walk subgraph sampling to provide the relative positional encoding for use with the transformer and which embedding to use in the subgraph transformer to use as the representation without incurring additional computational cost.

The only complaint is that some of the discussed related work isn't compared in the text and some of the baseline methods used are not discussed in the related work section either. This seems to cause a slight inconsistency between the two.

**Questions:**

(1) Why is the bidirectional GRU used instead of say pre-trained embedding models?

**Reviewer Confidence:**

3: The reviewer is confident but not certain that the evaluation is correct

**Scope:**

4: The work is relevant to the Web and to the track, and is of broad interest to the community

---

### Official Review · Reviewer_PirY · 2023-11-24

**Novelty:** 4
**Technical Quality:** 5

**Review:**

## Summary:
The authors propose a novel method for fake news classification that combines news articles, textual features, topics, and entity annotations in a heterogeneous network repreentation. A classifier is trained to recognize strucural outliers in this network representation and utilize them for fake news classification. The performance of the proposed method is demonstrated to improve over the SOTA on several benchmark data sets.

## Overall recommendation:
On the upside, the paper is well written, the proposed method is novel in design and provides good performance, and the provided experimentation is extensive. On the downside, the paper is incremental in a well researched task and the premise behind atypical subgraph-based classification is not sufficiently explored. Overall, the novelty is limited, but the paper would be a suitable fit to the program.

## Strengths:
* S1 By focusing on the classification of mis/disinformation, the paper addresses a timely and relevant topic
* S2 The proposed model utilizes a novel approach that the authors demonstrate to provide SOTA performance.
* S3 The paper includes exensive experiments, incl. ablative testing
* S4 The model description is extensive and a code repository is included, providing excellent reproducibility

## Weaknesses:

W1 Veracity of the central premise / assumption

I have concerns regarding the central premise that the authors use to motivate their approach, namely the focus on atypical graph structures. While I agree that some fake news is likely going to results in atypical graph structures, I am not convinced that this is necessarily always the case. In particular, would one not expect this approach to fail spectacularly in the case of long running propaganda campaigns? Conversely, some of the most influential true breaking news stories will also cause atypical patterns by linking hitherto unconnected entities. It would be good to perform an initial analysis into this premise, as well as an error analysis to demonstrate that the method does not falsely classify the most important real news alongside fake news. Alternatively, it would be good to explore the overlap of correctly classified examples with regard to prior baselines to establish the novelty.

W2 Model components

While the model description itself is extensive, little attention is given to some of the key components. Why is such an old tool like LDA used to determine topics when we have newer alternatives? I can see how the performance might still be good enough or even better, but it would be good to see this tested empirically. It would also be good to have an estimate of error propagation for entity detection and its impact on the proposed model.

W3 Incrementality

While the model itself is new, it is pushing performance on a well-researched task. Overall, despite SOTA performance, the paper is incremental in method development (i.e., the contribution boils down to how one can include even more features/information as input for a transformer arcitecture) and in performance as compared to existing baselines.

## Minor remarks:
* In line 135, "identifying and matching these subgraphs rely on the investigation of the heterogeneous graph, which is NP-hard" is quite vague. Can you be more concrete?
* In line 485 "we take the advantage of" -> "we take advantage of the"
* Figure 3 is quite difficult to read

## Post review update:
Increased novelty score (3 -> 4) after author feedback

**Questions:**

Q1: In case you have performed further investigation into the suitability of using atypical graphs for fake news classification beyond measurin the model's performance, could you please share them?

**Ethics Review Description:**

-

**Reviewer Confidence:**

3: The reviewer is confident but not certain that the evaluation is correct

**Scope:**

3: The work is somewhat relevant to the Web and to the track, and is of narrow interest to a sub-community

---

### Official Review · Reviewer_5pb4 · 2023-11-24

**Novelty:** 4
**Technical Quality:** 5

**Review:**

This paper proposes HeteroSGT, a heterogeneous subgraph transformer that captures both the textual features (sentence-level and word-level) and the structure for fake news detection. They first apply a dual-attention LM to derive textual features and then employ random walks with restart to extract subgraphs centered on each news. The extracted subgraphs are further fed to the proposed subgraph transformer for encoding and detecting the presence of fake news. Extensive experiments show the effectiveness of HeteroSGT over 5 baselines and on 5 datasets.

1. The paper is well-motivated and easy to understand.
2. It tackles an important problem of fake news detection and proposes HeteroSGT, a heterogeneous subgraph transformer that captures both the textual features and the graph structure for fake news detection.
3. It conducts comprehensive experiments and ablation studies on 5 datasets demonstrating the effectiveness of the approach.

Usually, random walk based approaches are time and memory intensive. It is worth comparing their approach with the SOTA approaches in terms of the training/inference time and the memory requirements.

“For MM COVID, the optimal walk length is 11, and the restart probability is 0.1” – did you use the same/different walk length and restart probability for the other 4 datasets? It’s important to report that in the paper.

From figure 5, it is difficult to infer which walk length and restart probability are effective as the line plots show zig-zag trends and for some walk lengths/restart probabilities, m-precision is higher with lower m-recall leading to low m-F1 score.


I appreciate authors' thorough rebuttal response and running additional experiments. Upon adding these additional technical details and experiments, the next version of the paper will be much stronger. My concerns have been addressed successfully. I believe the current ratings give justice to the work.

**Questions:**

“For MM COVID, the optimal walk length is 11, and the restart probability is 0.1” – did you use the same/different walk length and restart probability for the other 4 datasets? It’s important to report that in the paper.

Random walk-based approaches are usually time and memory intensive. What do you think? Is it worth comparing the proposed approach with the SOTA approaches in terms of the training/inference time and the memory requirements?

**Reviewer Confidence:**

4: The reviewer is certain that the evaluation is correct and very familiar with the relevant literature

**Scope:**

4: The work is relevant to the Web and to the track, and is of broad interest to the community

---

### Official Review · Reviewer_ny21 · 2023-11-27

**Novelty:** 4
**Technical Quality:** 4

**Review:**

This paper proposes a heterogeneous subgraph transformer (HeteroSGT) to detect fake news, which integrates pre-trained dual-attention module to get node representation and random walks to get subraph sequence. Here are some pros and cons of this paper:
Pros:
1.	This paper proposes a novel heterogeneous subgraph transformer (HeteroSGT) - to exploit subgraphs in constructed heterogeneous graph for fake news detection.
2.	This paper also performs a comprehensive results analysis and ablation study to show the effectiveness of HeteroSGT.
However, this paper also has some cons:
1.	It seems that edges are not considered in the feature learning process. How to link the entities and topics with different edges and how to learn representations via BERT are not clearly justified.
2.	Why not use BERT or Sentence-BERT in 3.3.1 and 3.3.2 rather than Bi-GRU? Since Bi-GRU is not the latest and most effective technology.
3.	It’s not clear why RWR is used in the METHODOLOGY section.
4.	Baseline settings are somewhat simple, missing some SOTA comparisons such as KAN, dEFEND.

**Questions:**

1.	RWR is essentially a random walk and why can it confirm position information? Will there not be a situation where a topic or entity at a later position in the sequence is more relevant to the news node? Assume that these entities are directly connected to news (line456).
2.	Why not use BERTopic rather than LDA?

**Reviewer Confidence:**

4: The reviewer is certain that the evaluation is correct and very familiar with the relevant literature

**Scope:**

3: The work is somewhat relevant to the Web and to the track, and is of narrow interest to a sub-community

---

### Decision · Program_Chairs · 2024-01-22

**Decision:**

Accept (Oral)

**Comment:**

Our decision is to accept. Please see the AC's review below and improve the work considering that and the reviewers' feedback for cemera-ready submission. We also ask that you more clearly state and defend assumptions around atypical fake news subgraphs in your camera-ready submission.

"This paper proposes a heterogeneous subgraph transformer that captures textual and structural features for fake news detection. They use a dual-attention LM to derive the textual features and then use random walks with restart to extract subgraphs centered on each news. The extracted subgraphs are further fed to the proposed subgraph transformer for encoding and detecting the presence of fake news. The experimental results show the effectiveness of the tool over several baselines and datasets.

 Strengths:

 - The paper address a relevant problem, fake news detection, with a novel approach
 - It performs extensive experiments and ablation studies on 5 baselines and datasets demonstrating the effectiveness of the approach.
 - The model is well described and includes a code repository, providing good reproducibility
 - The paper is well-motivated and easy to read.

 Weaknesses:
 - The research is based on the strong assumption that all fake news will have atypical graph structures and that true news will not have them. In general this will not be the case and hence more work is needed to understand the impact of false negatives and positives.
 - The components of the model can be improved as there are newer alternatives. Why those were not used?
 - In spite of the combination of techniques, the improvements are incremental

 The rebuttal generated new results that should be included in the final version if the paper is accepted.

 Scope: 4; Novelty: 5; Quality: 4"